# Apathy-Related Symptoms Appear Early in Parkinson’s Disease

**DOI:** 10.3390/healthcare10010091

**Published:** 2022-01-04

**Authors:** Emmie Cohen, Allison A. Bay, Liang Ni, Madeleine E. Hackney

**Affiliations:** 1College of Arts and Sciences, Emory University, Atlanta, GA 30322, USA; emmie.taylor.cohen@emory.edu; 2Department of Medicine, Division of Geriatrics and Gerontology, School of Medicine, Emory University, 1841 Clifton Rd., Atlanta, GA 30329, USA; allison.bay@emory.edu (A.A.B.); liangni12@gmail.com (L.N.); 3Atlanta VA Center for Visual and Neurocognitive Rehabilitation, Decatur, GA 30033, USA; 4School of Nursing, Emory University, Atlanta, GA 30329, USA; 5Birmingham/Atlanta VA Geriatric Research Education and Clinical Center, Birmingham, AL 35233, USA; 6Department of Rehabilitation Medicine, School of Medicine, Emory University, Atlanta, GA 30329, USA

**Keywords:** Parkinson’s disease, apathy, Hoehn and Yahr Scale, quality-of-life, non-motor symptoms, neuropsychiatric symptoms

## Abstract

Background: Apathy, often-unrecognized in Parkinson’s Disease (PD), adversely impacts quality-of-life (QOL) and may increase with disease severity. Identifying apathy early can aid treatment and enhance prognoses. Whether feelings related to apathy (e.g., loss of pleasure) are present in mild PD and how apathy and related feelings increase with disease severity is unknown. Methods: 120 individuals (M age: 69.0 ± 8.2 y) with mild (stages 1–2, *n* = 71) and moderate (stages 2.5–4; *n* = 49) PD were assessed for apathy and apathy-related constructs including loss of pleasure, energy, interest in people or activities, and sex. Correlations were used to determine the association of apathy with apathy-related constructs. Regression models, adjusted for age, cognitive status, and transportation, compared groups for prevalence of apathy and apathy-related feelings. Results: Apathy-related constructs and apathy were significantly correlated. Apathy was present in one in five participants with mild PD and doubled in participants with moderate PD. Except for loss of energy, apathy-related constructs were observed in mild PD at a prevalence of 41% or greater. Strong associations were noted between all apathy-related constructs and greater disease severity. After adjustment for transportation status serving as a proxy for independence, stage of disease remained significant only for loss of pleasure and loss of energy. Conclusion: People with mild PD showed signs of apathy and apathy-related feelings. Loss of pleasure and energy are apathy-related feelings impacted by disease severity. Clinicians should consider evaluating for feelings related to apathy to enhance early diagnosis in individuals who might otherwise not exhibit psychopathology.

## 1. Introduction

Parkinson’s Disease (PD), the second most common neurodegenerative disease in the United States, involves dopamine depletion in the substantia nigra and ventral tegmental areas [1]. Reduced dopamine levels lead to Parkinson’s motor symptoms and can result in the neuropsychiatric symptom, apathy. Characterized by loss of motivation, decreased activity, reduced enthusiasm, decreased interest, initiative, emotional indifference, a lack of concern, [2] and decreased motivation [3], apathy results in anhedonia (i.e., loss of pleasure) [4,5]. Traditionally, researchers have suggested a possible role of cognitive mechanisms in the expression of apathy. Apathy in people with PD is multidimensional and caused by dysfunction from different neural systems [2]. Impairments of distinct prefrontal cortex-basal ganglia circuits establish apathy subtypes. Apathy arises from a lesion and/or disruption of the prefrontal cortex-basal ganglia axis, which aids to control self-generated purposeful behavior [6,7]. Prefrontal cortex lesions are often observed in neurodegenerative diseases, such as, PD as well [8]. In early stages, some evidence shows apathy may be responsive to Levodopa treatment [9]. However, over time and even with gold-standard levodopa treatments, apathy is an often-intractable symptom of PD [8]. Currently, no strong level 1 evidence exists for treatment of apathy [10]. Early identification of apathy, especially in individuals with de novo PD, is vital to manage PD [11], given apathy results in decreased quality-of-life (QOL) [12] and may be a precursor or predictor of more severe disease [10] and cognitive impairment [13,14].

Previous studies demonstrate the prevalence of apathy in PD to vary between 16.5% [15] and 42% [16]. Although previous studies reported apathy in people with mild PD at a prevalence of 32% [17,18], currently, little information exists regarding the relationship between apathy and increasing disease severity. In a study using the Movement Disorders Society Unified Parkinson Disease Rating Scale (MDS-UPDRS) Part I (self-report of motor experiences of daily living) item 1.5 to measure apathy in PD, 104 out of 241 participants with mild-moderate PD (stages I–III) had apathy [19].

While neurologists address, consider and study akinesia and motor function, they may sometimes underestimate the importance of evaluating and treating apathy due to disease progression. More knowledge about the association between apathy and disease severity staging in PD would better allow clinicians to treat apathy symptoms and fine-tune participant-centered treatment of non-motor symptoms (NMS). No study has evaluated apathy comparing mild stages to moderate stages to determine the effect of increased disease severity. Further, psychological constructs that are related to apathy include fatigue, depression, and symptoms or feelings associated with depression, e.g., loss of pleasure, interest in people or activities, energy, and interest in sex [20]. Apathy may present alone or in concomitance with depression [6]. Apathy is strongly associated with depression, thus differentiating these syndromes remains challenging. As reported previously, up to 50% of people with PD who also have apathy exhibit concomitant depression, which is more prevalent in individuals with greater disease severity [13]. While apathy is not a clinical criterion of depression, it may be a clinical expression of a depressed state. Apathy is not the consequence of depression in most neurological diseases [21,22]. Assessing apathy-related constructs associated with depression could be useful in identifying indicators of apathy early in the PD process [10] by improving neuropsychiatric symptom treatment in earlier stages.

This study aimed to determine the association between apathy level, per the MDS-UPDRS part I, [19] and stage of PD (early stages grouped into “mild” versus later stages grouped into “moderate” categories) in individuals from a large metro area in the southeastern United States. Given that apathy is likely related to dopamine depletion [23], we hypothesized that the percentage of participants with moderate PD (Hoehn and Yahr (HY) stages 2.5, and 3) with some apathy would exceed the percentage of those with mild PD (HY stages 1, 1.5 and 2). Secondly, we wanted to determine the prevalence of feelings similar to apathy, which could enhance the early detection of apathy in PD. Therefore, we first investigated the validity of apathy-related constructs from the Beck Depression Questionnaire-II (i.e., the loss of pleasure, the loss of interest, the loss of energy, and the loss of interest in sex) with a Parkinson metric of apathy, the MDS-UPDRS part I item 1.5. Next, we compared mild and moderate groups on the prevalence of these apathy-related constructs. In analyses, we covaried for age, cognitive status and transportation status because of these covariates’ strong relationship to worse neuropsychiatric symptoms [24,25,26]. Given that loss of pleasure, energy, interest in people and activities and interest in sex are possibly direct causes of or direct results of apathetic feelings [1,8], we hypothesized that the apathy-related constructs and apathy are strongly associated. We hypothesized a higher prevalence of apathy-related constructs than apathy itself in participants with mild PD. Because of the dopamine depletion associated with increasing disease severity [23], after adjustment for the important confounders mentioned above, this association would strengthen with increased disease stages for all apathy and apathy-related feelings.

## 2. Materials and Methods

The Institutional Review Board at Emory University School of Medicine and the R&D Committee of the Atlanta VA approved this work. Participants provided written informed consent before participating (IRB00080676; IRB00060613).

### 2.1. Participants

120 participants with Parkinson’s Disease (PD) participated in the study. We recruited participants through the United States Department of Veterans Affairs (VA) registry, the VA Informatics, and Computing Infrastructure (VINCI) database, the Michael J. Fox finder website, the Movement Disorders Unit of Emory University, PD organizations’ newsletters, support groups, and educational events and through word of mouth.

Participant Inclusion criteria follow. They were aged 40 years and older, had received a clinical diagnosis of PD from a Movement Disorders Society trained movement disorders neurologist, using standard, Movement Disorders society criteria. This diagnosis indicates that at the time of diagnosis eligible participants reported unilateral onset of symptoms, exhibited two of three cardinal signs (tremor, rigidity, and bradykinesia), and showed clear symptomatic benefits from antiparkinsonian medications, e.g., levodopa [27]. Participants were required to be able to walk at least three meters unassisted and were classified into HY stages I–IV [28]. Exclusion criteria follow: We excluded participants who reported major neurological disorders other than PD (e.g., previous stroke, traumatic brain injury, or peripheral neuropathies). Additionally, no one with untreated moderate or major depression could participate.

The scale of PD stage measurement is the Hoehn and Yahr Scale (HY), developed in 1967. The scale includes five stages and was recently adjusted to account for intermediate stages, adding stages 1.5 and 2.5 [29]. Stage 1 has unilateral involvement (minimal disability), Stage 1.5 has unilateral and axial involvement, stage 2 has bilateral involvement without balance impairment, stage 2.5 has mild bilateral disease with recovery on pull test, stage 3 has mild-moderate disability, impaired posture, and bilateral involvement, stage 4 represents those who are severely disabled, but these individuals still can stand unassisted, and stage 5 includes individuals with bed confinement [30]. Participants were grouped by disease severity, with HY ≤ 2.5 being classified as “mild” and HY > 2.5 classified as “moderate.” We determined these groupings because participants belonging to the “mild” group are considered to have no postural instability, while those in the group, “moderate” have postural instability. Postural instability is a strong indicator of an individual’s level of independence, which is related to apathy and other neuropsychiatric symptoms.

### 2.2. Assessment

We used standard, valid and reliable tests in the assessments. Participants were administered a health questionnaire a week before their appointment. Medication status, including anti-parkinsonian medication, was gathered through this questionnaire and verified in an interview at the visit. At the appointment, participants were characterized for general ability to complete activities of daily living (ADLs) with the Composite Physical Function Index (CPF) [31,32]. Participants were tested while ON (i.e., participants were medicated with antiparkinsonian, (e.g., levodopa), treatment) for all assessments except for the Movement Disorders Society Unified Parkinson Disease Rating Scale part III (MDS-UPDRS-III), the rated motor exam, during which they were tested in the OFF (participants are off treatment) state, i.e., at least 12 h after having taken their last anti-parkinsonian medication. All participants were being treated by a movement disorders neurologist and were following a prescribed medication regimen. MDS-UPDRS is the recommended scale of the Movement Disorders Society for rating symptom severity in PD. Part I (self-report of motor experiences of daily living), II (self-report of non-motor experiences of daily living), III (observer rated motor exam), and IV (medication-related motor fluctuations- interview) [33] were administered. The reliability of the MDS-UPDRS part I is supported as an outcome measure in PD studies [34]. Parts I and III were administered in the study. Part I assesses the non-motor impact of PD on patients’ experiences of daily living. There are 13 questions. Part 1A is administered by a rater (six questions) and focuses on complex behaviors.

#### Covariates

Age, cognitive status (as measured by the Montreal Cognitive Assessment (MoCA)), and transportation status (dichotomously determined as either transportation that was self-provided or the reliance on others for transportation) were used as important covariates in regression models to determine the relationship of disease stage (mild or moderate) with apathy and other outcome variables. These covariates were chosen for their strong association with neuropsychiatric symptoms. Previous studies reported an association between increased age, greater neuropsychiatric symptoms, impaired cognitive function and worsened quality-of-life (QOL) [24,25,35,36]. Individuals with apathy have been shown to be on average 3.3 years older than those without apathy [9]. Other reports demonstrated similar significant relationships between apathy and lower cognitive function [3,8,15,16,24,37]. Transportation status was chosen as a proxy for independence and because the examination of baseline clinical characteristics and demographics revealed differences between groups on this important covariate.

### 2.3. Assessment of Apathy and Depression

No agreed-upon gold standard for assessing apathy in PD exists [38,39]. Further validation must be included when a gold standard for assessing apathy is defined [2,40]. We used the apathy item derived from the MDS-UPDRS (1.5). The MDS-UPDRS is correlated with validated rating scales of QOL and functional disability [41]. There is a relationship between the MDS-UPDRS apathy construct and executive functioning, dopamine dysregulation syndrome, and cognitive impairments [42]. In the apathy item derived from the MDS-UPDRS interview section of Part I (question 1.5), the interviewer observes the level of spontaneous activity, motivation, assertiveness, and initiative and then rates the impact of reduced levels of performance in daily life and social interaction. The single item, Apathy (1.5), has been called an “easy to assess and suitable” item for screening for apathy and has been shown to have validity with the longer Lille Apathy Rating Scale, rho = 0.516, *p* < 0.001 [43].

We also analyzed the fatigue item from the MDS-UPDRS (1.13) because of its highly correlated relationship with apathy [44,45,46]. These items are rated on a Likert scale from 0–4, with zero signifying no symptoms and four being most severe. We used items from the Beck Depression Inventory-II (BDI-II) to determine the levels of apathy-related constructs. The BDI-II consists of 21 items related to depressive symptoms of how the participant is feeling over the previous two weeks. We performed targeted analyses focusing on the items that correlated most with apathy, which include the loss of the following: pleasure (BDI-II.4), interest (BDI-II.12), energy (BDI-II.15) and interest of sex (BDI-II.21). A higher score indicates more depression [47]. The cut-off for participants with moderate depression on the BDI-II is 18 [48]. Both the MDS-UPDRS and the BDI-II are not limited by floor and ceiling effects because psychometric studies showed fewer than 15% of all participants achieved highest or lowest scores [34,49].

Participants were also administered the Geriatric Depression Scale, as well as the complete BDI-II scale.

### 2.4. Neuropsychological Assessment

Cognitive assessments included the Montreal Cognitive Assessment (MoCA), a measure of global cognition [50].

### 2.5. Statistical Analysis

Data were entered and cross-checked for quality assurance. The probability of having apathy in mild and moderate disease stages was estimated [17,18,19], and these probabilities were used to estimate the expected prevalence of apathy in our own sample between mild and moderate stage participants. Our analysis revealed that our sample of 120 individuals has nearly 80% (77%) power to detect differences in the proportion of individuals with apathy between mild and moderate stage PD groups at the 95% confidence level. We acknowledge this 77% does not reach 80% power which is generally considered acceptable to avoid type 2 error. Therefore, findings from this pilot study should be interpreted very cautiously. Future work is necessary to confirm the findings presented here.

We binned participants into the following categories of PD: HY stages 1–2 (Mild) and 2.5–4 (Moderate). To characterize the sample, we calculated descriptive statistics, including levodopa equivalency dosage for a subset of participants for whom data were available, and using the method as published by Tomlinson et al. (2010) [51]. We analyzed demographic and clinical characteristics for differences between groups using independent *t*-tests and chi-square tests as appropriate. Spearman correlations assessed associations between the MDS-UPDRS Part I and BDI-II variables.

Because of the skewed distribution of responses with high frequencies of zero responses across items, we dichotomized the variables: Apathy (MDS-UPDRS 1.5), Loss of Pleasure (BDI-II.4), Loss of Interest in People or Activities (BDI-II.12), and Loss of Interest in Sex (BDI-II.21) into variables with two categories (0: “none”, >0: “some”). In the variable without high frequencies of zeros: Loss of Energy (BDI-II.15), we dichotomized according to a reasonable distribution of high versus low on the scale into two groups (0–1: “No/Slight Loss”; 2–3: “Mild/Severe Loss”). We dichotomized the fatigue (MDS-UPDRS 1.13) variable into two categories (0–1: “Normal/Slight/Mild”; 2–4: “Moderate/Severe”).

To identify how group status (level of disease severity) affected levels on the apathy/fatigue-related constructs, we used multivariate logistic regression models for the dichotomous variables. Model 1 adjusted for age, model 2 adjusted for age and MoCA, and model 3 adjusted for age, MoCA, and transportation. Transportation was adjusted because of the distribution noted in the groups. The independent variables used in our models included group, age, MoCA, and transportation. The dependent variables were the dichotomized BDI-II items. The odds ratios (OR) ±95% confidence interval (CI) expressed the association between the apathy-related constructs and the severity of HY stages. We performed analyses using R (R Core Team 2017, Vienna, Austria).

## 3. Results

The 120 individuals with Parkinson’s Disease (PD) were classified into Hoehn and Yahr Scale (HY) stages 1–2, “mild,” *n* = 71 (stage 1, *n* = 7, stage 1.5, *n* = 15, stage 2, *n* = 49); or HY stages 2.5–4 (stage 2.5, *n* = 17, stage 3, *n* = 29, stage 4, *n* = 3), “moderate,” *n* = 49. Table 1 presents the participant characteristics. The sample was 25% non-white, and for 26 participants who reported Levodopa dosage, the sample took an estimated 616 mg of L-dopa daily. The moderate HY group was older, and more at risk for loss of function, had PD longer, had higher BDI-II scores, (but not GDS scores), had lower MoCA scores, were slightly more likely to live in senior living, were less likely to transport themselves, left their house less frequently, took more prescribed medications and were more likely to use an assistive device than the mild group. BDI-II values are presented in Table 1 ranging from M = 11.4–15.5 (8.5, 8.7), indicating mild depression among some participants. GDS scores are also presented.

Correlations between apathy-related items.

Table 2 presents the associations between the scores of apathy-related constructs on the MDS-UPDRS Part I and the BDI-II in a correlation matrix. We noted significant, moderate positive correlations between apathy, loss of pleasure, loss of interest in people or activities, and loss of interest in sex.

Apathy items.

Table 3 presents the associations between apathy/apathy-related constructs and PD staging along with the corresponding OR. Models 2 and 3 are discussed here. Approximately 21% of participants with mild PD reported apathy, and 44% of participants with moderate PD reported apathy. Model 2 (adjusted for age and MoCA) shows the odds of reporting apathy to be 3.74 times greater among moderate participants than participants with mild PD, and that in Model 3 (adjusted for age, MoCA, and transportation) the odds fell to 2.47 times greater and were not significant after controlling for transportation status. Moreover, the analyses demonstrated that transportation status removed the significance of disease stage on most of the related feelings.

Approximately half of the sample in both groups reported fatigue, and no models were significant for effect of PD stage. Fifty-four percent of participants with mild PD and 78% of participants with moderate PD reported loss of pleasure. The odds of reporting loss of pleasure were 5.11 times greater among participants with moderate PD than participants with mild PD after controlling for both age and MoCA (Model 2), remaining significant after controlling for transportation in model 3.

Forty-one percent of participants with mild PD and 57% of participants with moderate PD reported loss of interest in people and activities. The odds of reporting loss of interest in people and activities were 2.14 times greater in participants with moderate PD than participants with mild PD after controlling for age and MoCA score (Model 2) but were not significant after transportation was added (Model 3).

Eleven percent of participants with mild PD and 22% of participants with moderate PD reported loss of energy. The odds of having loss of energy were 3.59 times greater among participants with moderate PD (Model 2), increasing to 5.06 times greater after controlling for age, MOCA and mode of transportation (Model 3).

Forty-seven percent of participants with mild PD and 65% of participants with moderate PD reported a loss of interest in sex. The odds of participants with moderate PD having greater loss of interest in sex than participants with mild PD were not significant in any of the models after controlling for age, MOCA and transportation (Table 3).

Appendix A presents mean and standard deviations for ordinal outcome variables.

## 4. Discussion

In this cohort, apathy prevalence in moderate stages (44%) was twice that of mild stages. All apathy-related constructs were strongly associated with apathy except for loss of energy. In contrast to apathy prevalence, in mild stages of PD, loss of pleasure, interest in people or activities, and sex were highly prevalent at >41%. Disease staging impacted apathy and related feelings, given that over 50% of the moderate group reported fatigue, loss of interest in people or activities, loss of interest in sex, and/or loss of pleasure. However, with all three covariates controlled for in Model 3, the premise that apathy increased with disease severity is not fully supported by our data, disagreeing with some previous work that did not control for these covariates [52]. After being added to the model, transportation status was an impactful covariate, given that stage of disease effect was no longer significant for apathy, loss of interest in people/activities, loss of interest in sex, and fatigue. In contrast, odds remained significant for loss of pleasure and loss of energy.

### 4.1. Apathy-Related Feelings and Early Disease Detection

A novel contribution of this work was demonstrating the construct validity of apathy-related constructs from a non-PD-specific metric, BDI-II, to apathy as measured by a PD-specific metric, MDS-UPDRS Part I, the apathy item. Adding to previous work, we found that loss of pleasure, interest in people or activities, and interest in sex were strongly correlated to apathy and fatigue [53,54,55]. All BDI-II items, except the loss of energy, were correlated to apathy within our study. Apathy-related constructs and feelings may help with the early detection of apathy and related sequelae. Apathy, not depression, is associated with increased cortical amyloidopathy in PD, agreeing with work showing a relationship with cognitive decline [56]. Cortical and striatal β-amyloidopathy has been previously noted in apathetic PD participants compared to non-apathetic PD participants [57]. Executive cognitive dysfunction is most apparent in apathetic PD participants when apathy has been evaluated and determined based on participant emotional, interest, and motivation responses [8]. Previous research reported a clear negative correlation between apathy and cognitive status. Impaired cognition has been associated with increased apathy at even early stages [52]. Therefore, apathy may be an early biomarker for cognitive impairments in PD [52] and must be identified as soon as possible.

### 4.2. High Prevalence of Apathy-Related Feelings in Mild Stages

The high prevalence of several apathy-related feelings in mild stages could be utilized as an indicator of likely development of apathy in moderate stages. Future research will further investigate the relationship between these feelings and the progression to apathy in moderate stages. Indications of related feelings would possibly serve as precursors to the neuropsychiatric symptom of apathy and its related symptomatology [58].

### 4.3. People with Moderate PD Reported Higher Odds of Experiencing Loss of Pleasure and Loss of Energy than People with Mild PD

Currently, the relationship between apathy and anhedonia is unclear in PD because the Diagnostic and Statistical Manuel of Mental Disorders-IV (DSM-IV) does not distinguish between them [59]. Previous work reported high anhedonia levels in apathetic compared to non-apathetic people with PD [8]. The high odds of loss of pleasure and loss of energy in our findings may be universal experiences in moderate stages, regardless of age, cognition, and transportation. The loss of energy most likely represents a symptom of depression [60], which is related to apathy but has its own pathological and clinical course.

### 4.4. Lack of Transportation and Independence May Contribute to Apathy

We found a relationship between disease severity and apathy using models with (a) no covariates, (b) controlling for age only and c) for both age and cognition, which is consistent with previous studies [3,24,44,61,62,63]. Interestingly, our analysis demonstrated that the inclusion of transportation status removed the significance of disease stage on apathy and most related feelings. Other work has also reported no relationship between disease severity and apathy, though this work did not examine transportation status as a proxy for independence [43]. Transportation is a fundamental human necessity for all ages and has been linked to autonomy, independence, and quality-of-life (QOL) [64,65,66]. Regarding health-related QOL, self-evaluated autonomy is a key predictor and important aspect of neurodegenerative disease course, for example, that of PD [67]. Autonomy is reduced by the physical and emotional burdens presented to the participant and their family during the disease course [68]. Driving, especially in the United States, is regularly linked to autonomy and symbolizes personal independence for adults [64,69]. A high level of cognition, motor functioning, and independent mobility is necessary to drive a vehicle [70]. Transportation is related to independence as it represents a basic human need to get from place A to B [65]. As such, the independence that is lost with inability to provide one’s own transportation may be highly associated with feelings of apathy and far more important to one’s mood than even the motor impairments experienced by the individual with moderate PD. Our findings indicate that independence is highly impactful on apathy and related feelings. While transportation dependence seems to increase with age, age may not be the ultimate explanation. Rather, poor driving-related abilities due to conditions such as PD may better explain patterns of transportation dependence [71]. Covarying for transportation status allowed us to remove variance related to participants’ state of independence. Individuals with limited independence must depend on caregivers and others for everyday functioning, which is greatly associated with decreased quality of life [72]. More research is necessary to investigate the relationship between independence in PD and reports of apathy.

### 4.5. Understanding Fatigue May Be Important in People with Apathetic PD

Our findings support a relationship between apathy and fatigue. Only about 1 in 10 participants reported a loss of energy in both groups, though more than half of the sample reported fatigue. The PD-centric metric of fatigue from the MDS-UPDRS was reported in approximately half of our sample, with early and late PD reporting roughly the same prevalence. These findings agree with previous work that showed fatigue present in 44% of the total PD population [73] and apathy and fatigue presence in all stages of PD [74,75].

### 4.6. Limitations

Limitations in this study include using the single item from the MDS-UPDRS as a measurement of apathy. Other apathy scales, such as the Starkstein Apathy Scale and Lille Apathy Rating Scale, are more accurate measures of apathy but were unavailable for this secondary data analysis. However, the MDS-UPDRS is the gold standard scale used by neurologists to evaluate people with PD. The apathy item within the MDS-UPDRS comes from the original motivation/initiative item in the UPDRS [76], and this item has been shown to have clinical utility for screening and high correlation with a more complete apathy scale [43] By dichotomizing the Apathy item, we can only determine if apathy is mildly or moderately present. Additional valid and standard measures of apathy should be included in apathy specific research moving forward. However, this work was concerned with the clinical utility of the scales used here, given the purpose of this work was to identify other items that will probe and screen for apathy in people with early PD. Akinesia levels were not examined in this study. Future work will examine the relationship between levels of bradykinesia specifically and reported apathy. Our findings should be interpreted with caution due to the large CI (95%) used due to the small sample size and high dispersion of data. We excluded participants with major neurological disorders, which limits generalizability. Larger sample sizes are necessary for future studies to be able to make more definitive conclusions. Further, very few participants (*n* = 2) were stage IV, and no participants were stage V; thus, no conclusions about severe PD and apathy can be made. We used transportation status as a proxy for independence, and because of observed differences between groups; however, other measures may be more suitable for measuring independence for future research examining the relationship between independence, disease severity and apathy.

## 5. Conclusions

More individuals with moderate PD than mild individuals report apathy; however, their disease level does not explain why they report apathy. Independence levels, age, and cognitive status may account for apathy more than the disease stage itself. While a relationship between disease severity and apathy/apathy-related constructs exists when controlling for age and cognition, most of the significant effects of the disease stage disappear after controlling for transportation. Our results contradict the postulation that apathy increases with disease severity, which has been the overall consensus from several previous studies that did not adjust for our covariates.

In our participants with mild PD, the data show the prevalence of apathy-related constructs of fatigue, loss of pleasure, loss of interest in people or activities, and loss of interest in sex at greater than 41%. This information is important because previous studies have presented data that posit apathy as a possible early expression of executive impairment and cognitive processing in PD [77]. Early identification of apathy, specifically in de novo PD participants, is essential for optimal clinical management [6]. Participants are less likely to report severe cognitive deficits in mild versus moderate stages of PD [26,78]. Identifying apathetic signs early in PD should receive greater attention from neurologists following these participants and drive more referrals to mental health specialists.

Several apathy-related neuropsychiatric symptoms may be detected in people with mild PD, while apathy itself appears to be reported by those with moderate PD more. The quality-of-life (QOL) of individuals with PD may be enhanced with the early detection of apathy [59]. Increasing the methods including specifically evaluating the participant for loss of energy and loss of pleasure, may help identify people with PD who are at risk of developing greater nonmotor symptoms in PD. Understanding the increased risks of apathetic neuropsychiatric symptoms in people with moderate PD will lead to enhanced preventative and palliative treatments. In addition, providers must understand and identify signs of apathy within mild stages of PD because apathy is a predictive factor for parkinsonian decline, and cognitive decline over time.

## Figures and Tables

**Table 1 healthcare-10-00091-t001:** Demographic characteristics of participants with mild vs. moderate stages of Parkinson’s disease.

		Whole Sample	Mild (1–2)	Moderate (2.5–4)	*p*
N		120	71	49	
Age		68.98 ± 8.2	66.89 ± 7.9	71.98 ± 7.8	**0.001**
H&Y, N (%) ^	1	7 (5.8)	7 (9.9)	0 (0)	**<0.001**
	1.5	15 (12.5)	15 (21.1)	0 (0)	
	2	49 (40.8)	49 (69)	0 (0)	
	2.5	17 (14.2)	0 (0)	17 (34.7)	
	3	29 (24.2)	0 (0)	29 (59.2)	
	4	3 (2.5)	0(0)	3 (6.1)	
		36.24 ± 13.23	29.73 ± 11.29	45.67 ± 9.73	**<0.001**
MDS-UPDRS III (score)					
Education (years)		16.27 ± 2.5	16.61 ± 2.6	15.69 ± 2.3	0.098
Body Mass Index (kg/m^2^)		26.59 ± 5.2	26.87 ± 5	26.18 ± 5.5	0.480
No. Comorbidities		3.45 ± 1.9	3.36 ± 1.9	3.58 ± 1.8	0.519
Composite Physical Function (/24) Number of Medication		18.25 ± 5.4 4.88 ± 3.62	20.7 ± 3.5 4.78 ± 2.93	14.62 ± 5.8 7.03 ± 4.33	**<0.001** **<0.001**
Time with PD (y)		6.96 ± 5	5.38 ± 3.7	9.24 ± 5.8	**<0.001**
MoCA Score (/30)		24.8 ± 4.2	26.3 ± 2.8	22.8 ± 5.0	**<0.001**
BDI-II total score Geriatric Depression Scale (GDS)		12.9 ± 8.7 4.15 ± 3.52	11.4 ± 8.7 3.87 ± 3.30	15.5 ± 8.5 4.61 ± 3.84	**0.010** 0.347
Sex, N (%) ^					0.982
	Men	77 (64.2)	45 (63.4)	32 (65.3)	
	Women	43 (35.8)	26 (36.6)	17 (34.7)	
Race (B/W), N (%) ^					0.471
	Black	22 (18.3)	12 (16.9)	10 (20.4)	
	Other	9 (7.5)	7 (9.9)	2 (4.1)	
	White	89 (74.2)	52 (73.2)	37 (75.5)	
House Type, N (%) ^					**0.020**
	Assisted/senior living	10 (8.4)	2 (2.8)	8 (16.7)	
	Self/independently	109 (91.6)	69 (97.2)	40 (83.3)	
Transportation, N (%) ^					**<0.001**
	Other	33 (27.7)	7 (9.9)	26 (54.2)	
	Self	86 (72.3)	64 (90.1)	22 (45.8)	
Leave House, N (%) ^					**<0.001**
	1–2 times/week	10 (8.5)	2 (2.8)	8 (17)	
	1 time/week	1 (0.8)	0 (0)	1 (2.1)	
	3–4 times/week	37 (31.4)	13 (18.3)	24 (51.1)	
	Daily	70 (59.3)	56 (78.9)	14 (29.8)	
Use of Assistive Device, N (%) ^					**<0.001**
	No	55 (46.6)	44 (62.9)	11 (22.9)	
	Yes	63 (53.4)	26 (37.1)	37 (77.1)	

Participant Baseline Characteristics. Continuous variables values are presented as Mean ± Standard Deviation. Categorical variable values are presented as N (%) denoted by “^.” To determine differences between the Mild and Moderate groups, *p* values were calculated with an independent *t*-test for continuous variables, and the Chi-square test was used for categorical variables. Bolded values are significant. Significance level was set at *p* < 0.05.

**Table 2 healthcare-10-00091-t002:** Correlations between apathy and apathy-related variables.

Items		Apathy (MDS-UPDRS 1.5)	Fatigue (MDS-UPDRS 1.13)	Loss of Pleasure (BDI-II.4)	Loss of Interest in People or Activities (BDI-II.12)	Loss of Energy (BDI-II.15)	Loss of Interest in Sex (BDI-II.21)
Apathy (MDS-UPDRS 1.5)	ρ	1 *	0.39 *	0.33 *	0.31 *	0.16	0.25 *
	*p*	0	0	0	0.001	0.084	0.006

* Values were Spearman correlation coefficients.

**Table 3 healthcare-10-00091-t003:** Comparison of outcome variables between the mild group and moderate group.

		Mild (1–2)	Mod. (2.5–4)	Unadj		Model 1		Model 2		Model 3	
**Items**		N (%)	N (%)	OR Moderate vs. Mild (95% CI)	*p*	OR Moderate vs. Mild (95% CI)	*p*	OR Moderate vs. Mild (95% CI)	*p*	OR Moderate vs. Mild (95% CI)	*p*
Apathy (MDS-UPDRS 1.5)	No apathy	56 (79)	27 (56)	2.90 (1.31, 6.61)	0.010	3.34 (1.43, 8.15)	0.006	3.74 (1.52, 9.6)	0.005	2.47 (0.91, 6.73)	0.075
	With apathy	15 (21)	21 (44)								
Fatigue (MDS-UPDRS 1.13)	Normal/Slight/Mild	38 (54)	19 (41)	1.69 (0.8, 3.62)	0.173	1.96 (0.89, 4.46)	0.1	1.96 (0.84, 4.69)	0.123	1.44 (0.57, 3.62)	0.438
	Moderate/Severe	32 (46)	27 (59)								
Loss of Pleasure (BDI-II.4)	No	33 (46)	11 (22)	3.0 (1.35, 7.02)	0.008	4.93 (2.02, 13.12)	0.001	5.11 (1.96, 14.76)	0.001	4.75 (1.72, 14.58)	0.004
	Yes	38 (54)	38 (78)								
Loss of Interest in People or Activities (BDI-II.12)	No	42 (59)	21 (43)	1.93 (0.93, 4.08)	0.080	2.30 (1.05, 5.17)	0.04	2.14 (0.93, 5.08)	0.077	2.34 (0.95, 5.98)	0.068
	Yes	29 (41)	28 (57)								
Loss of Energy (BDI-II.15)	No/Slight loss	63 (89)	38 (78)	2.28 (0.85, 6.37)	0.105	3.15 (1.09, 9.75)	0.038	3.59 (1.17, 11.69)	0.027	5.06 (1.54, 17.95)	0.009
	Mild/severe loss	8 (11)	11 (22)								
Loss of Interest in Sex (BDI-II.21)	No	37 (53)	17 (35)	2.11 (1, 4.54)	0.052	2.40 (1.09, 5.47)	0.033	2.19 (0.95, 5.26)	0.071	1.73 (0.7, 4.37)	0.235
	Yes	33 (47)	32 (65)								

Unadjusted (Unadj): ORs (odds ratios) and *p* values were obtained from the raw logistic regression model. Model 1: Logistic regression model adjusting for age. Model 2: Logistic regression model adjusting for age and MOCA score. Model 3: Logistic regression model adjusting for age, MOCA score, and Transportation. Mod. = Moderate.

## Data Availability

The data that support the findings of this study are available on request from the corresponding author. The data are not publicly available due to privacy or ethical restrictions.

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
