# Peer review of "Apathy-Related Symptoms Appear Early in Parkinson’s Disease"

_healthcare, 2022, doi:10.3390/healthcare10010091_

Round 1

Reviewer 1 Report

In the manuscript “Apathy-related Symptoms Appear Early in PD”, Cohen et al. investigate whether patients with mild Parkinson’s disease (PD) experience feelings associated with apathy and how prevalence of apathy and related feelings change with disease severity. Therefore, apathy and associated emotions were determined in 120 patients with mild or moderate PD. Furthermore, the correlation of apathy with apathy-related feelings was analyzed. To adjust for confounding variables such as age, independence level and cognitive status, regression models were used. Although apathy itself was twice as prevalent in PD patients with moderate disease compared to patients with mild disease, about 40% of the latter showed apathy-related feelings. The authors observed a strong association between apathy-related constructs and disease severity and find that transportation status is a major factor influencing the experience of some apathy-related feelings. Overall, they conclude that age, cognitive status and independence level rather than disease stage per se determine the presence of apathy in PD patients. Moreover, they suggest that early detection of apathy-related emotions can help to identify patients at risk to develop more severe nonmotor symptoms, thus allowing early intervention potentially enhancing life quality.

Broad comments

Addressing apathy in PD, the study approaches an important topic which fits well into the scope of Healthcare, especially when considering that apathy and related feelings not only significantly impact the patient, but also poses an enormous burden for family members and caregivers. Thus, the reported findings can contribute to the understanding of apathy in PD and improve the patients’ quality of life. Overall, the manuscript benefits from appropriate use of language, an informative introduction and a detailed discussion which also addresses some limitations of the study. However, there are some weaknesses outlined below:

1) Apathy is strongly associated with depression, thus differentiating these syndroms still remains challenging. As reported previously, up to 50% of PD patients with apathy exhibit concomitant depression, which is more prevalent in patients with greater disease severity. Therefore, to investigate “pure” apathy and to be able to draw meaningful conclusions regarding this syndrome, it would be very important at least to include information about depression status in the demographic characteristics of the patient groups, or even better, to exclude patients with concomitant depression.

2) To evaluate the presence of apathy, the MDS-UPDRS scale was used in the study. Although this metric can be used to screen for apathy in PD patients, it is not recommended for diagnosis, and more accurate measures, such as the Starkstein Apathy Scale and the Lille Apathy Rating Scale, are available for research purposes. Combining these tools would facilitate a reliable diagnosis. However, the authors shortly acknowledge this pitfall in their limitation section and point out that additional measures of apathy must be included in further research.

3) Other limitations, which are also addressed in the limitations section, are the small size of the study cohort and the lack of data for patients with severe disease.

Specific comments

1) To make the significant findings of Table 1 more obvious to the reader, it would be nice if the respective p-values could be highlighted, for example by writing them in bold, as done in Table 2 and 3.

2) Although overall, the manuscript is well-written, there are some language points that could be improved:

2.1 It would be beneficial to introduce abbreviations or medical terms before their first use, for example:

  • Title: PD: I would recommend using the full term (Parkinson’s Disease)
  • Line 45: QOL
  • Line 123: ON (patients are on treatment)

2.2 Terms should be written in a consistent style, for example apathy-related (line 18) vs. apathy related (line 28).

2.3 Some sentences are difficult to follow. Editing these sentences would significantly help the reader to understand the findings, for example line 84-87. Furthermore, in some instances the wording is inappropriate, for example instead of using “mild” or “moderate” participants (line 212, 316), it would be better to refer to “participants/patients with mild/moderate disease”. Similarly, the heading of Table 1 should be adapted (Demographic characteristics of patients with mild vs. moderate stage of PD)

Author Response

Please see the attachment, thank you!

Reviewer 2 Report

In general the study addresses an important issue. The study has some major methodological flwas. Apathy is defined here from only a single item (which is also dichotomized). This item is then correlated with - theoretically not justified - items of the BDI and corrected for a few cofactors. It turns out, as expected, that BDI items and apathy are related and that apathy increases with increasing HY stage. However, the main cofactors are not included (depression, LEED, medication, antidepressant therapy, etc.). Therefore, the study is subject to circular reasoning and the validity of the apathy construct is not sufficiently proven. It would have been better to prospectively design the study better (with an appropriate apathy score which could then be validated in PD), instead of retrospectively selling any correlations from a data set as valid constructs. 

Introduction:

  • "While neurologists are very focused on akinesia and motor function" Please provide evidence and reference for this statement. Many neurologists are aware of NMS in PD.
  • Please detail the definition and construct of apathy in general.
  • Please provide rationale for using selected BDI-items as "apathy construct". Please provide rationale for selection of items. What apathy related aspects are missing in your "apathy construct".
  • Please discuss the overlap with depression.
  • "We hypothesized apathy-related constructs would be prevalent in mild PD at greater prevalence than apathy itself" Given that your apathy construct is derived from a depression score (depression and subthreshold depression are highly prevalent in PS), this hypothesis is not surprising.

Methods:

  • Please provide sample size calculation.
  • The enrolment is not clear (where, when, who). Provide detailed data about screening, inclusion and exclusion. Use a CONSORT diagram.
  • Were people with depression included?
  • Why do you use Brain Bank criteria instead of the MDS criteria?
  • "excluded participants who reported neurological disorders" please detail; this means that also patients with headache or PD patients with polyneuropathy were excluded. This limits generalisability.
  • How dis you assure that assessments week before their appointment were done in the On-State?
  • "Transportation status was chosen as a proxy for independence." A valid ADL score would have been a better choice.
  • Dichotomizing ordinal values means that you loose information. Given that your main measure (apathy) is only derived from one(!) MDS-UPDRS item, I do not think it is reasonable to dichotomize this apathy item. Then you cannot determine levels of apathy, you only can make a statement if apathy is present or not with low validity.
  • Table 1. You mean the MDS-UPDRS III and not the UPDRS III, right?
  • Table 2. Here you correlate the dichotomized apathy item (yes/no) with the other dichotomized items?
  • Please provide detailed data for the ordinal items (as supplement)
  • Sample size seems to low for the number of statistical test. This leads to high risk of false positive results.
  • report floor and ceiling effects
  • consider to use backward selection instead of nested models

Discussion

  • "demonstrating the construct validity of apathy-related constructs from a non-PD-specific metric" You only demonstrated moderate correlations.
  • why was loss of energy not correlated with apathy?

Author Response

Please see the attachment, thank you!

Reviewer 3 Report

The authors investigated the association between apathy and apathy-related constructs: pleasure, energy, interest in people or activities, and sex. The study involved a large cohort of 120 patients with Parkinson’s Disease (PD) with mild and moderate stages. They authors found that: 1/ apathy-related constructs and apathy were significantly correlated, 2/ apathy was present in one in five participants with mild PD and doubled in participants with moderate PD, 3/  loss of pleasure and energy are apathy related feelings impacted by disease severity.

The paper is well written, clear and well supported. Moreover, considering the association between apathy and apathy-related is an original and interesting viewpoint.

However, I would recommend the authors to take into account the following comments to improve the quality and the robustness of the manuscript.

Major comments

Comment 1.  Apathy

Apathy is a key concept in this paper. However, the authors provide description of apathy, relying on its features (loss of motivation, decreased activity, reduced enthusiasm, decreased interest, initiative, emotional indifference, a lack of concern and decreased motivation) without mentioning the multifactorial nature of apathy and its subtypes based on the impairment of distinct prefrontal cortex-basal ganglia circuits. Traditionally, researchers identified: 1/ emotional/motivational apathy, i.e., lack of concern and limbic affective input as reward sensitivity; 2/ cognitive apathy, i.e., absence of initiated behavior due to executive dysfunction as planning; and 3/  behavioral/auto-initiation apathy, i.e., diminished self-initiated actions. Moreover, according Pluck & Brown (2002), “apathy in Parkinson’s disease can be distinguished from other psychiatric symptoms and personality features that are associated with the disease, and it is closely associated with cognitive impairment. These findings point to a possible role of cognitive mechanisms in the expression of apathy”. In their review, Pagonabarraga et al. (2015), reported that apathy in patients with Parkinson's disease is multidimensional and caused by dysfunction in different neural systems.

The authors should highlight apathy as a multifaceted construct and add a short literature review to identify the current state of art regarding the forms of apathy in PD.

Comment 2.  Assessment of apathy

  • It is unclear why the term “PSYCHOSOCIAL” was added to characterize the “ASSESSMENT OF APATHY”.

  • The authors said “No agreed-upon gold standard for assessing apathy in PD exists” referring to Lopez and al. (2019). However, Lopez and al. (2019) point rather “the inability of the AS (Starkstein Apathy Scale) to capture distinct cognitive and behavioral apathy factors in PD raises questions about the distinction between these dimensions in this disease”.

Lopez F V., Eglit GML, Schiehser DM, et al. (2019). Factor Analysis of the Apathy Scale in Parkinson’s Disease. Mov Disord Clin Pract. 432;6(5):379-386. doi:10.1002/mdc3.12767

  • It is unclear why the reference [27] was used to describe the Composite Physical Function Index (CPF).

[27] Dong X, Chang E-S, Simon MA. Physical Function Assessment in a Community-Dwelling Population of U.S. Chinese Older 419 Adults. Journals Gerontol Ser A Biol Sci Med Sci. 2014;69(Suppl 2):S31-S38. doi:10.1093/gerona/glu205

The authors should refer to Siu, Reuben, and Hays (1990) and explain in a more detailed way what is the Composite Physical Function (CPF).

Siu, A. L., Reuben, D. B., & Hays, R. D. (1990). Hierarchical measures of physical function in ambulatory geriatrics. Journal of the American Geriatrics Society, 38(10), 1113–1119.

Comment 3.  STATISTICAL ANALYSIS and RESULTS

The authors used three models: 1/ Model 1 adjusted for age, 2/ Model 2 adjusted for age and MoCA, and 3/ Model 3 adjusted for age, MoCA, and transportation.

The authors said that “21% of mild participants reported apathy, and 44% of moderate participants reported apathy”,  and that in Model 2 (adjusted for age and MoCA),”the odds of reporting apathy were 3.74 times greater among moderate than mild participants” and that in Model 3 (adjusted for age, MoCA, and transportation) “the odds fell to 2.74 times greater and were not significant after controlling for transportation status”. Moreover, the analyses demonstrated that transportation status removed the significance of disease stage on most of the related feelings”.

  • Unfortunately, the Table 3 shows “2.47” rather than “2.74”, regarding apathy. What is the correct values amongst these one.
  • The fact that “transportation status” impacts the results is a very interesting point, and the authors should further discuss this.

Minor comments

Comment 4.  Abbreviations

The QOL acronym was used without be spelled out in line 45.

The MDS-UPDRS Part I acronym was used in line 52 but not explained as “nonmotor experiences of daily living” until line 127.

The HY acronym was used in line 70 but not explained as “Hoehn and Yahr Scale” until line 104.

Author Response

Please see the attachment, thank you!

Round 2

Reviewer 2 Report

Also in the revision the major issues remain. The revision is also not convincing, as many aspects were not adequately addressed.

They should examine their apathy construct by doing factor analysis and not by providing background information in the intro.

Important measures and parameters are still missing: depression, LEDD

In the revision they suggest that this study was exploratory (the try to use this argument to justify their low sample size); however, then they cannot make confirmatory statements.

"We acknowledge this 77% does not reach 80% power which is generally considered acceptable. Therefore, findings from this study should be interpreted very cautiously". You should better try to reach the necessary sample size and then submit a new paper.

Geriatric Depression Scale scores is not mentioned in the methods.

In the models they have to correct for depression, use of antidepressive drugs and LEDD. To say that they do not want to change the selection mode (backward is definitively better that nested models) in their model is also not a sufficient answer in a revision procress.

This paper remains circular reasoning.  Apathy is defined only from a single item. It not validated and no construct validity is provided. This item is then correlated with -theoretically not justified and not validated -items of the BDI and corrected for a few cofactors. It turns out, as expected, that BDI items and apathy are related,and that apathy increases with increasing HY stage.